# OpenReview forum: "VoiceAgentBench: Are Voice Assistants Ready For Agentic Tasks?"
_TMLR — Rejected by TMLR_

### Review · Reviewer_Jzhe · 2026-03-17

**Summary Of Contributions:**

This paper introduces VoiceAgentBench (VAB), a benchmark comprising synthetic spoken queries across English and six Indic languages (Hindi, Bengali, Marathi, Tamil, Telugu, Malayalam) for evaluating speech language models on agentic tool-use tasks.
The benchmark covers six evaluation categories of increasing complexity: single tool invocation, single tool with retrieval, parallel tool calling, sequential dependent tool calling, multi-turn dialogue-based tool invocation, and safety evaluation.
 The authors evaluate three end-to-end SpeechLMs (KimiAudio 7B, Qwen2.5-Omni 7B, AudioFlamingo3 7B) and three ASR-LLM pipelines (Whisper v3 + Qwen3 8B / Gemma3 27B / Llama3.3 70B). The main findings are: (1) ASR-LLM pipelines consistently outperform end-to-end SpeechLMs; (2) performance degrades sharply on Indic languages; (3) sequential dependent tool calling is the hardest category; and (4) safety refusal rates are low and inconsistent, especially for SpeechLMs.

Key strengths: (a) The multilingual Indic coverage filling a genuine gap, no other benchmark evaluates speech-based agentic tool-use across six Indic languages. (b) The task taxonomy is well-designed, progressively isolating capabilities from simple parameter filling to complex dependency chains and multi-turn tool calling. (c) The evaluation framework with layered metrics (TS, TCS, PF, RR) provides interpretable failure mode analysis. (d) The ablation studies are informative.

Key weaknesses: (a) The paper's central claim of being "the first benchmark specifically designed to evaluate agentic tool-use capabilities in speech-based setting" is overstated and inadequately qualified, multiple prior and concurrent works explore spoken tool-use evaluation that are either dismissed in a single sentence or entirely omitted. (b) The related work section is dismissive of a growing body of work on speech-native tool use and retrieval, including SpeechRAG (Min et al., ICASSP 2025), SEAL (Sun et al., 2025), end-to-end RAG for S2S dialogue (Feng et al., EMNLP 2025 Findings), Stream RAG (Arora et al., 2025) (c) The paper does not evaluate the model's ability to use retrieved results from tools to improve final response generation. (d) The sequential dependent tool calling subset is small (80 English examples).

**Audience:**

Yes

**Audience Explanation:**

Voice-based AI agents represent a rapidly growing area at the intersection of speech processing and agentic AI, with significant commercial interest (GPT-4o voice, Gemini Live, Alexa LLM).
The paper addresses a timely question and the findings are of direct interest to researchers working on speech language models, LLM agents, multilingual NLP, and AI safety.
The multilingual Indic focus is particularly valuable given the scale of the Indian market for voice assistants and the scarcity of evaluation resources for these languages in agentic settings.
The benchmark itself, once released, would serve as a practical evaluation tool.

**Broader Impact Concerns:**

No major concern

**Claims And Evidence:**

No

**Claims Explanation:**

While the experimental methodology is generally sound and many of the empirical findings are well-supported, one of the paper's central claims, that VoiceAgentBench is "the first benchmark specifically designed to evaluate agentic tool-use capabilities in speech-based setting", is not adequately supported and requires significant qualification. This inaccuracy in positioning, combined with a dismissive treatment of closely related work, undermines the paper's contribution framing.

**1. The "first" claim is overstated and the related work is dismissive of prior art.**

This is my primary concern. The paper acknowledges concurrent work by Tan et al. (2025), Arora et al. (2025), and Wu et al. (2025) only in a single dismissive sentence at the end of Section 2: This one-liner does not do justice to the scope of prior work on speech-based tool use and retrieval:

- **SpeechRAG** (Min et al., arXiv 2412.16500,  ICASSP 2025)

- **SEAL** (Sun et al., arXiv 2502.02603, January 2025)

- **End-to-end RAG for S2S dialogue** (Feng et al., arXiv 2505.00028, April 2025, EMNLP 2025 Findings) presented a full end-to-end RAG framework for speech-to-speech dialogue systems, with evaluations on HotpotQA and RGB-zh — a multilingual benchmark for speech-based tool integration.

- **Stream RAG** (Arora et al., arXiv 2510.02044, October 2025) presents the first framework for integrating streaming tool use into speech-in speech-out systems. It additionally introduces AudioCRAG with both TTS synthesized and human spoken queries.

- **AURA** (arXiv 2506.23049)

- **BFCL Audio** (Mao, Ginart et al., Salesforce/Berkeley) directly evaluates function-calling accuracy from spoken input, comparing pipeline vs. end-to-end architectures, this is the most closely overlapping concurrent benchmark and is entirely absent from the paper.

 A more accurate and honest positioning would be to claim VoiceAgentBench as the first **multilingual, multi-turn** benchmark for speech-based tool use, while properly acknowledging and comparing against the substantial body of prior work on English speech tool-use evaluation. Table 1 should include BFCL Audio, and the Stream RAG's AudioCRAG to give readers a fair picture.

**2. ASR-LLM pipelines outperform end-to-end SpeechLMs (well-supported, but with caveats).**

This finding is robustly supported by Tables 3-5. However, the model selection systematically excludes SpeechLMs that were explicitly trained for tool-use. Step-Audio (StepFun AI, February 2025) and Step-Audio 2 (July 2025) are end-to-end SpeechLMs with explicit tool-calling support; LUCY (Tencent, January 2025) was fine-tuned for function calling. Excluding all tool-use-trained SpeechLMs while concluding that "SpeechLMs lag behind ASR-LLM pipelines" overstates the architectural gap, as the tested SpeechLMs were simply never trained for this capability.

**Requested Changes:**

1. **Substantially revise the related work section and retract the unqualified "first" claim.** The current treatment of prior work on speech-based tool use is dismissive to the point of being misleading. Please read my comment on whether the claims are supported.

2. The authors should include at least one tool-use-trained SpeechLM (Step-Audio, LUCY).

---

### Review · Reviewer_ZjUj · 2026-03-18

**Summary Of Contributions:**

This paper proposes VoiceAgentBench, a new benchmark for evaluating speech language models in spoken, agentic settings across English and six Indic languages. It builds on existing English text-based agent benchmarks, provides translated versions in six Indic languages, and generates corresponding speech data using TTS systems. The authors evaluate three speech LMs and three ASR+text LM cascade systems, demonstrating the strong performance of cascade approaches and highlighting the challenges of multilingual (Indic language) settings. The paper is clearly written and easy to follow. The primary contribution is the introduction of this benchmark. All artifacts will be publicly released, ensuring strong reproducibility and making the work valuable for researchers developing voice agents.

**Additional Comments:**

- Consider including proprietary systems in the evaluation, as they may offer stronger performance and provide additional insights.
- Clarify the justification for model choices, e.g., why Gemma3 27B and GPT-4o-mini are used for query generation for a different purpose?
- For Tables 3 and 4, please check whether the reported precision is appropriate. Given the relatively small sample size (on the order of hundreds, as in Table 2), reporting percentages to two decimal places may not be statistically meaningful.

**Audience:**

No

**Audience Explanation:**

I found the overall contribution and findings to be of limited value for the following reasons:

As I mentioned above, the benchmark is created by converting text-based agent tasks into speech via TTS, which significantly limits realism. As a result, the empirical findings are largely expected. Since the underlying tasks are fundamentally text-based, it is unsurprising that ASR+text LM cascade systems outperform speech LMs. Potential advantages of speech-native models, such as leveraging paralinguistic cues, are not meaningfully evaluated in this setup.

Similarly, the multilingual results are not particularly surprising, as performance differences largely reflect the current limitations of models in these languages rather than new insights. Finally, the multilingual scope is relatively narrow, focusing only on a set of Indic languages.

Overall, while the topic is relevant, the current findings may not substantially advance understanding for a broad portion of TMLR’s audience.

**Broader Impact Concerns:**

The authors’ ethical statement adequately addresses speech data considerations, including PII concerns, and appropriately notes that IRB approval is not required due to the use of TTS-generated data.

**Claims And Evidence:**

No

**Claims Explanation:**

The benchmark is constructed by converting existing text-based agent benchmarks into speech using TTS systems, which limits its realism. As a result, the claim that it reflects realistic voice agent scenarios is somewhat overstated. In particular, truly realistic voice interactions involve rich paralinguistic cues, such as emotion, speaker characteristics, and prosody, which are not adequately captured in this setup. Consequently, the evidence does not fully support the claims about realism.

**Requested Changes:**

- Improve the realism of the benchmark by incorporating actual recorded conversations or by carefully repurposing spoken dialogue data to better reflect spoken interactions.
- Include tasks that explicitly require speech-specific (paralinguistic) information, such as speaker identity, emotion, and prosody.
- Expand the multilingual coverage beyond Indic languages to include a broader and more diverse set of languages.

---

### Review · Reviewer_QX7M · 2026-03-26

**Summary Of Contributions:**

Authors propose VoiceAgentBench which evaluates SpeechLMs and ASR-LLMs for agentic tasks. They create a benchmark of 6000+ queries across multiple language and tool use cases including adversarial defense. On their synthetic VAB, they show how ASR-LLMs dominate across all variations. Results show the gap between the two model classes. Choice of sub-models is also latest and results have good insights. Paper is written well.

**Audience:**

No

**Audience Explanation:**

At this stage, it is more of speech processing case study paper to find the gap between SpeechLMs and ASR-LLM on instruction following, which can be measured through much simpler experiments. I suspect the breath is low for TMLR audience.

**Claims And Evidence:**

No

**Claims Explanation:**

Good effort, but it did not accomplish any of the research directions it endeavored.
Weaknesses:
1. Benchmark is fully synthetic. SpeechLMs would probably be more suitable for natural audio (noisy, prosodic variations, etc.). SpeechLMs were set up for failure in current methodology. Perhaps thats ok and authors wanted to measure the performance gap between SpeechLMs and ASR-LLMs but if that's the case, the paper becomes audio processing paper rather than more generic agentic eval paper. Same gap can be measured with much simpler experiments.
2. Indic direction is weak. Perhaps Kannada language could have been included. In any case, then multi-turn cases should be included for all languages, otherwise the benchmark is unfair to some languages. Also, "cultural" part is not defined which is repeatedly claimed. What unique properties in your regional language queries exist which are not in English?
3. Agent direction is weak. I think agents must be able to do lot more.

I think paper tried to hit many spots. I doubt the community will use the current version of VAB for benchmarking audio agents due to above mentioned limitations.

**Requested Changes:**

Paper has structural weaknesses. If it had claimed less, it would have been ok but to propose a benchmark for whole community to use, it would need strong corrections in all 3 weakness i mentioned: 1) Synthetic queries, 2) Weak basis for claiming cultural diversity, 3) Simplistic agentic tasks.

---

### Decision · Action_Editor_HmW4 · 2026-04-23

**Recommendation:** Reject

**Audience:**

Yes

**Audience Explanation:**

Strong agentic benchmark for spoken agents will be interested for wide audience as it expands research from text-based LLM to speechLLMs in agentic world.

**Claims And Evidence:**

No

**Claims Explanation:**

**Paper summary**

The paper proposes a new benchmark for speech models to evaluate agentic setting. Benchmark spans 6k+ queries across English and sic Indic languages. Data in benchmark are fully synthetic. Also perform speechLLMs and cascaded models evaluation, showing that cascaded models outperform e2e speechLLMs.

**How claims are supported**

While reviewers pointed that paper is well written, topic is critical and timely discussed, choice of sub-models is also latest and results have good insights, **there are several critical issues with the paper**:
- All reviewers pointed out that benchmark is fully synthetic, which significantly limits realism. "As a result, the empirical findings are largely expected." Paralinguistic role is not evaluated.
- (Reviewers QX7M, ZjUj) There are limitations in multilinguality: scope is relatively narrow, focusing only on a set of Indic languages, and multi-turn cases should be included for all languages, otherwise the benchmark is unfair to some languages. Also, "cultural" part is not defined which is repeatedly claimed.
- Agentic tasks are very simple.
- (Reviewer Jzhe) Claim "the first benchmark specifically designed to evaluate agentic tool-use capabilities in speech-based setting" is incorrect as there many prior work exists. Related works are not adequate and the work is not properly placed within prior work. Thus  novelty claim is overstated (though novelty is not a criterion for TLMR).
- (Reviewer Jzhe) Authors exclude from comparison speechLLMs which are specifically trained for tool-use, thus current benchmarking doesn't reflect full picture.

**Discussion / revision**

Authors did not participated in discussion and did not provide any revision based on reviewers comments.